# Experiences and perspectives on chimeric antigen receptor (CAR) T-cell therapy among recipients, carers and referrers (RE-TELL): a qualitative study to inform CAR T-cell service design

Robert Fyfe  ,[1,2] Olivia Anstis,[3] Kushant Kapadia,[3] Mallory Jordan,[3] Danielle Oriwa Sword,[1] Robert Weinkove[1,4]

This study was presented in a provisional form at the Blood Conference 2022.

¹Malaghan Institute of Medical Research, Wellington, New Zealand
²Te Herenga Waka - Victoria University of Wellington, Wellington, New Zealand
³Health Advisory, Deloitte Limited, Auckland, New Zealand
⁴Te Rerenga Ora Blood & Cancer Centre, Te Whatu Ora Health New Zealand Capital Coast and Hutt Valley, Wellington, New Zealand

**Correspondence to**
Dr Robert Fyfe;
robert.fyfe@ccdhb.org.nz

## ABSTRACT

**Objectives** RE-TELL is a qualitative study, which aims to understand patient, support person, clinician and coordinator experiences and perspectives of chimeric antigen receptor (CAR) T-cell therapy, to inform design of a clinical CAR T-cell service in Aotearoa New Zealand.

**Design** Semistructured qualitative interviews focused on domains of: experience through treatment, elements that work well and those that could be improved on. Interviews used thematic analysis to identify key themes. A workshop was held to obtain participants' reflections on interim analysis and proposed improvements.

**Participants** New Zealanders with experience of CAR T-cell therapy, including recipients, support persons, clinicians and coordinators.

**Results** We interviewed 19 participants comprising 5 CAR T-cell recipients, 3 support persons, 6 clinicians and 5 coordinators. Four participants identified as Māori. Thematic analysis identified three global themes. The first, 'sociocultural factors impact CAR T access', identified potential sources of inequity including geographic, financial and informed consent barriers. The second, 'varying emotions, roles and enablers', identified an easier treatment experience compared with alternatives; an underwhelming cell administration process; frustration with inpatient monitoring; burden on support persons and importance of 'bridge' organisations such as charities and patient support groups. Lastly, 'golden opportunities: reimagining CAR T service delivery', suggested: improved geographical access to CAR T-cell therapy, while retaining consolidated clinician experience; a 'dashboard' with information on CAR T-cell treatment, time frames and manufacture; a health navigator to co-ordinate non-medical aspects of treatment and signpost care; embedding of indigenous data sovereignty and ownership of cells; a cell infusion ceremony, incorporating family involvement and Māori cultural elements and outpatient administration and monitoring where possible.

**Conclusion** This study documented the current experience of New Zealanders receiving CAR T-cell therapy and identified opportunities for future service development. These insights are relevant to service design within Aotearoa New Zealand, and other countries developing equitable CAR T-cell services.

## STRENGTHS AND LIMITATIONS OF THIS STUDY

⇒ Diverse viewpoints pertaining to CAR T-cell therapy in Aotearoa New Zealand.
⇒ Many facets of treatment experience explored.
⇒ Specific focus on strategies to improve indigenous Māori health outcomes.
⇒ Data saturation may not have been reached for all areas.
⇒ Further research is required to develop more solutions to existing challenges.

## INTRODUCTION

Chimeric antigen receptor (CAR) T-cell therapy has become a standard of care for some relapsed and refractory B-cell malignancies.[1 2] As a personalised cell and gene therapy involving gene modification of a patient's own T-cells, access to treatment is limited by logistical challenges and cost, while varied concerns regarding the technology need to be addressed, including equity of access, optimising the patient and support person experience, and cell and data sovereignty.[3–5] Internationally, CAR T-cell therapy provision has not consistently ensured equity of access,[6 7] and equity has been highlighted as a key area of focus for ongoing development of this treatment modality.[8]

Aotearoa New Zealand (NZ) does not yet have a national service for standard-of-care CAR T-cell delivery, with access limited to clinical trial use[9] or overseas treatment. Within Aotearoa NZ, cancer-specific mortality is higher for indigenous Māori than for non-Māori, including for lymphoma and myeloma.[10] As a result, there is a strong focus on designing healthcare services to rectify this inequity.[11]

While the challenges of delivering equitable access to CAR T-cell therapy have been considered worldwide,[12 13] the Aotearoa NZ healthcare system has unique characteristics, such as an imperative to promote indigenous health due to the government's obligations to Te Tiriti o Waitangi/the Treaty of Waitangi (the nation's founding document, which guarantees Māori the right to access to equitable healthcare and health outcomes); low public pharmaceutical funding compared with some high-income countries[14] and wide geographical dispersal.[15]

We aimed to assess the current experiences of CAR T-cell therapy for patients, support persons, clinicians and administrators within Aotearoa NZ, to develop insights into treatment experience, needs and barriers to treatment; and to identify potential improvements for future services. Our overall goal was to inform design of a future national CAR T-cell service. We conducted semistructured qualitative interviews, assessed these interviews with thematic analysis, and further refined this analysis through a codesign workshop with participants, where the interim results of the thematic analysis were presented to participants for comment.

## METHODS
### Study design
In the RE-TELL study, we recruited a total of 19 participants, including 5 patients, 3 support persons, 6 clinicians and 5 administrators, to provide a diversity of perspectives on CAR T-cell therapy. At the time of recruitment, there were an estimated 11 NZ patients who had received CAR T-cell therapy, given the limitations of research into a novel therapy for non-Hodgkin's lymphoma. Of these, seven were invited to participate (and accepted), with the goal of recruiting five participants to provide a range of perspectives. Two patients then became unable to participate due to health or logistical reasons. We also intended to recruit five support persons, clinicians and administrators, to provide additional perspectives while not prejudicing the patient voice. Participants were selected using purposive sampling to encompass a range of experiences (treated in Aotearoa NZ or overseas; different outcomes after treatment; different roles in a patient's care) and a diversity of backgrounds (age, ethnicity, gender, urban and rural). They were contacted by the primary investigator by email with an invitation to participate and general information regarding the study.

A codesign methodology was chosen to take a partnership approach with patients and support persons, to understand patient experience, and to use this to develop potential improvements. Following the methodology of Boyd et al,[16] the project followed the six main elements of engage, plan, explore, develop, decide and change. Engage refers to establishing relationships with patients and staff to improve health services. Plan involves generating ideas about the goals of improvement work and strategies to achieve it. Explore refers to understanding patient and staff experiences, to identify areas for improvement. Develop involves turning ideas into specific improvements. Decide involves a choice between improvements and how to implement them. Change means then turning ideas into action.

### Data collection and analysis
Using semistructured interviewing techniques, we guided participants along the treatment pathway, enquiring of treatment experience, elements that worked well, elements that could be improved on, practical needs, informational needs, emotional experiences, cultural elements and system flow. With a deductive approach in mind, this aimed to provide a logical framework with which to structure themes and experiences, and for participants to review themes and work towards service improvements. The full interview guide is provided as online supplemental appendix 1. Interview data were analysed using thematic analysis, with transcripts themed using the above-mentioned theoretical/deductive approach. Following the framework provided by Braun and Clarke,[17] the following six steps were taken to analyse the data:

1. Familiarisation with the data: All interviews and the focus group were transcribed and independently reviewed by RF and OA. Participants were sent their own transcripts prior to analysis, so that they were able to read through and make corrections or add clarifications. The transcripts were referred to repeatedly throughout the analysis to understand and clarify meaning and resultant themes.
2. Initial coding: Labels and tags were assigned within the transcripts, to identify information that related to phases of the treatment journey.
3. Generating initial themes: Coding was independently organised into practical, emotional or cultural themes across the treatment pathway. New or novel information was highlighted to discuss between the researchers in the next phase.
4. Reviewing and Defining themes. RF and OA presented their coding and initial themes to each other, for agreement on global themes and subthemes.
5. Data charting: Themes were mapped across the treatment pathway into multiple streams in a chart highlighting the experience of clinicians, patients and whānau.
6. Reviewing and refining themes: Resultant themes were presented to participants to ratify and clarify in a focus group, and as a basis to form discussion on service improvements. These are the themes presented in this report.

All interviews and the workshop were facilitated by a health psychologist (OA) registered with the NZ Psychologist's Board. Interviews were performed between 13 January 2022 and 8 April 2022. Interviews were conducted via videoconference, as COVID-19 pandemic-related restrictions precluded in-person interviews. Interviews were audiorecorded,and lasted between 28 and 60 min.

Interviewees were given the option to review manuscripts and provide comments.

A focus group was held to review and refine themes with participants (n=7) who self-selected to attend a videoconference workshop, in order to gain reflections on the responses and potential service improvements. During this workshop, a graphical representation of the patient journey through CAR T-cell therapy was displayed, as described by participants of the RE-TELL study (online supplemental appendix 2). This served to summarise information gathered during the interviews, to provide logical structure to the themes by mapping them across the treatment timeline, and to help overcome health literacy barriers. The responses from workshop participants assisted with the formulation of 'table 1, Global theme 3: future improvements to CAR T-cell pathway' and the improvements emphasised in the discussion; no new themes were identified during the workshop. The primary themes from this workshop are presented in the Results and Codesign workshop section.

Interviews were transcribed verbatim and analysed using data analysis software NVivo V.11. Transcripts were analysed and coded independently into key themes by RF and OA according to existing guidelines.[17 18] OA had no prior involvement with the participants or CAR T-cell therapy; RF was the clinical fellow for the ENABLE trial during the study, with clinical involvement with two patients and one participant. The researchers then met to seek alignment and generate global and subthemes collaboratively. Subthemes were allocated to provide contextual explanation of key themes, or describe less common yet important topics or emotional descriptors. Themes were formulated into a table including sample belief statements and sample quotes.

The original study protocol is available as online supplemental file 3.

## Patient and public involvement

Patients were not involved in setting the research question or formation of the study protocol, but their responses directed the interviews and provided the themes. We intend to disseminate the results to study participants in a newsletter suitable for a non-specialist audience.

## RESULTS
### Baseline characteristics

We enrolled 19 participants, including 5 patients, 3 support persons, 6 clinicians and 5 administrators. Overall, 4 participants were Māori, 11 NZ European, 2 Asian and 1 other; 12 of 19 participants (63%) were female, 7 male. The highest level of education was secondary education for 4 participants and tertiary education for 15.

Among the five patients interviewed, all had B-cell non-Hodgkin's lymphoma and had received CAR T-cell therapy. Their median age was 48 years (range 23–59). Three had received investigational third-generation CAR T-cells within the phase 1 'ENABLE' trial in Aotearoa NZ[9]; one had received standard-of-care axicabtagene ciloleucel (axi-cel) in Australia via a government funding mechanism; and one had received lisocabtagene autoleucel in the USA within a clinical trial. Four of the five patients had a complete response to CAR T-cell therapy at the time of interview.

Of the three support persons, one had supported a patient through treatment via the ENABLE trial; one through treatment in Australia and one through treatment in the USA.

The clinicians interviewed comprised four clinical haematologists who had referred patients for CAR T-cell therapy, one palliative care nurse practitioner and one research nurse. The administrators comprised a charitable services coordinator, a bone marrow transplant coordinator, a transport coordinator and two social workers.

### Data saturation

Data saturation occurs when no additional themes emerge from the data. This was not reached in this study for all areas of analysis, and reasons for this are explored in the Discussion section. Specifically, many of the subthemes mentioned in global theme 3 did not reach saturation, as there were a multitude of potential areas improvements raised, which were not discussed with all participants.

### Interviews: key domains and themes

Themes arising from the interviews were organised into three global themes, before subclassification into 8 domains and 23 subthemes (see table 1). The first global theme, sociocultural factors impact CAR T access, relates to both the challenges that patients and clinicians face in attempting to access CAR T-cell therapy, and the barriers and inequities that may prevent others from accessing the treatment. The second global theme, varying emotions, roles and enablers, describes the experience of treatment within NZ or overseas, including the ancillary services in place to support patients. The final global theme, golden opportunities: re-imagining CAR T service delivery, explores the ways that a future national CAR T-cell therapy service might improve treatment delivery.

### Global theme 1: accessing CAR T-cell therapy
#### Potential inequity stemming from lack of access

Current mechanisms of access to CAR T-cell therapy in Aotearoa NZ are via clinical trial, self-funded treatment overseas or government-funded treatment overseas. There was broad agreement among those interviewed that current access to CAR T-cell therapy is inequitable, with the impression that those able to access treatment are likely to be more health literate, have more financial resources, are more likely to reside in an urban centre and exist within societal structures that enable their health literacy and ability to navigate the health system.

#### Requirement for self-advocacy

Noting the barriers to accessing CAR T-cell therapy identified above, a frequently raised theme was that patients and support persons were required to advocate for

**Table 1** Global themes, key domains, subthemes and sample quotes

| Relevant domains | Subtheme | Sample quotes |
|---|---|---|
| **Global theme 1: sociocultural factors impact CAR T access** | | |
| Equity | Potential inequity stemming from lack of access | 'To be equitable, you need to make sure it is accessible to everyone, and we can't just rely on clinical trials and compassionate access programmes' (C1). |
| | Geographical barriers | 'The first barrier is distance' (C1)<br>'I understand you can't have CAR T in all centres. I just think one is probably not enough for many patients' (C1)<br>'You are asking someone who has a limited support network to fly… it's hard for the patient psychologically' (C1) |
| | Informed consent | 'As soon as you start talking about how risky things are—a Māori worldview is often quality over quantity…. In a medicolegal worldview, we go into the nth degree about every small little potential risk, such that(Māori)will start to get scared and go well, actually, I don't want this.' (C5)<br>'You should give me information that I can just hand to my mother and she understands it' (P1) |
| | Financial barriers | 'People just don't have enough money… for financial support, transitioning from being a full time employee to having to stop.' (A4)<br>'We say to patients we have this intensive treatment—it's fully funded, all public… but we have no mechanism to help you stay down here, and you've got to a charity [for accommodation]' (A2) |
| | Awareness of trial | 'At least a couple of patients initiated the conversation about wanting to be part of the CAR T-cell trial' (C4)<br>'People don't know where to look sometimes' (C6) |
| | Requirement for advocacy | 'Our health system requires you to navigate it yourself, it's so patient unfriendly' (P1) |
| **Global theme 2: varying emotions, roles and enablers** | | |
| Patient experience | Underwhelming administration of cells | 'It's just a shot in your arm and then you're done… anti-climactic' (P5) |
| | Treatment experience easier compared with chemotherapy | 'It was quite chill… chemo's a hell of a lot worse' (P5)<br>'A walk in the park' (P2)<br>'It was actually the first time through the whole cancer journey that I got to sleep a lot… I just rested' (P3) |
| | Frustration with inpatient monitoring | 'I was so bored I went out to strip the bed, got some new sheets and made my bed up' (P2) |
| Support person experience | Emotional burden | 'The relief to get him to the hospital and know that somebody else was looking after him, and it wasn't me that was having to keep things going every day' (S3)<br>'There's no care for the carers' (S1) |
| | Requirement to provide care for patients | 'I felt like I'd gone from being a wife to a carer' (S3) |
| | Role in advocacy for patients | 'Half the time I don't think [the patient] knew what day it was' (S3) |
| Bridges | Charities | 'Sometimes patients and families do like to talk to an organization that understands, but is separate to the hospital… so they won't be judged' (A1)<br>'There is a period of time there where the hospital finishes their treating, and they may not be so unwell that a hospice is involved, but they're in that middle zone and it's really stressful.' (A1) |
| | National Travel Assistance programme | 'It's not meant to cover [all travel costs), but it's meant to … eliminate the barrier of those upfront costs' (A5) |
| | Patient support groups | 'Show respect for what they've gone through and being able to articulate that experience to others' (A1) |
| **Global theme 3: golden opportunities: reimagining CAR T service delivery** | | |

Continued

**Table 1** Continued

| Relevant domains | Subtheme | Sample quotes |
|---|---|---|
| Centralised point of information | Accessible prepared resources | 'It's good to have it delivered through a video. It's a bit easier to comprehend' (P1) 'Knowledge is power… if you've got the information and know what to expect, it takes a lot of the fear out of the process' (P3) |
| | Health navigator | 'Someone who's thinking about the whole patient's life and experience going through the process, not just the medical side of it… a generalist, and then the specialists are doing their work' (P1) |
| Alterations to treatment delivery | Outpatient therapy | 'I could have done a lot of that as an outpatient' (P2) |
| | Ceremony for cell infusion | Having family members, and potentially referring clinicians present for cell infusion: 'It will show [patients] how important it is' (C5) |
| | Need for appropriate accommodation | 'Really inappropriate [to require family members to sleep on the floor]' (P3) |
| Service design | Clinical staff experience | 'You need to have [dedicated] staff because it's a unique therapy…(bone marrow transplant co-ordinators) couldn't just absorb(a CAR T-cell therapy role), they'd need extra resource' (A2) |
| Indigenous rights | Data sovereignty | 'Is it information for information's sake, or is there a purpose at the end?' (C5) |
| | Cell sovereignty | 'They told us what they were going to do with the stem cells if they didn't need them… whereas in Australia? We never got those options' (S2) 'You almost want a courier tracking system [for cells]' (C5) |

themselves, either to access CAR T-cell therapy in the first instance, or to shape their clinical care to fit their needs. Although referrals for CAR T-cell therapy occurred via a patient's haematologist/oncologist, several patients had to raise the option of CAR T-cell therapy with their treating clinician before this was considered. Many felt that the in the absence of standard-of-care CAR T-cell therapy in Aotearoa NZ, knowledge of referral pathways relies on 'word-of-mouth' communication.

### Geographical barriers
Aotearoa NZ is a country with a low population density, long transport times between centres and is distant from international centres that offer CAR T-cell therapies. Geographical inequity was identified as a factor impacting patients' ability to access therapy. Geographical barriers impacted people's finances and ability to care for their children and resulted in time away from support networks and whānau (family), and concerns about housing and job security.

### Informed consent
It was noted that many aspects of accessing treatments in Aotearoa NZ require an ability to navigate a complex medical system. In particular, the current process of consent for investigational treatments requires extensive cataloguing of possible risks of therapy. Participants commented that if this is provided without an equivalent explanation of potential benefits, patients may leave with the understanding that CAR T-cell therapies are riskier than they truly are (rather than accepting that the Western medicolegal paradigm requires robust documentation of each risk) (see table 1: global theme 2). While this issue with informed consent was noted by both NZ

European and Māori patients, it was thought that it may have a greater impact on Māori patients. Interviewees noted that this could be further exacerbated by differences in health literacy and an inability for many clinicians to provide information within a Te Ao Māori (Māori world) framework, which could contribute to an erosion of trust in the health system.

### Financial barriers
Although patients attempting to access CAR T via clinical trial in Aotearoa NZ experienced barriers, these were even more formidable for patients attempting to self-fund CAR T-cell therapy overseas. Among those accessing overseas treatment, patients and family were required to identify overseas hospitals with actively enrolling CAR T-cell therapy programmes; liaise with these hospitals and arrange transfer of medical information; fundraise high sums through combinations of personal funds, fundraising events or crowdfunding; and co-ordinate logistics to arrive at the treating hospital—all while unwell with their disease. Unsurprisingly, it was noted that many who considered exploring treatment via this route were eventually unable to access this.

### Global theme 2: varying emotions, roles and enablers
Treatment experience easier than chemotherapy: The predominant theme regarding the treatment experience was the relative tolerability of CAR T-cell therapy. New Zealanders who accessed CAR T-cells within the ENABLE trial did so as a last-line therapy, so had experience of numerous chemotherapy cycles, and often stem cell transplantion. Participants reported that, although there were adverse effects, CAR T-cell therapy was much more comfortable and tolerable than their prior treatments.

Indeed, the primary difficulty with CAR T-cell therapy was reported to be the prolonged inpatient monitoring (typically 14 days), or (for the recipient of axi-cel in Australia) monitoring in an intensive care unit for cytokine release syndrome, while feeling well. Additionally, many of the treatment challenges highlighted were not primarily attributable to physical adverse events, but rather to socioeconomic factors. For those treated in Aotearoa NZ, these included accessing suitable accommodation for the patient and support person; affording groceries and compensating for loss of income while a support person relocated to accompany the patient. For those treated outside Aotearoa NZ, these included identifying overseas treating centres and securing a treatment place; complex and expensive logistical planning and balancing having a support person at the treating centre to support the patient with maintaining family life at home. It was commented that conducting follow-up in an outpatient setting would have overcome many of the challenges of treatment (see global theme 3).

### Underwhelming administration of cells

It was noted that the infusion of CAR T-cells could be anticlimactic, since it involved a very brief cell infusion. While some patients found the moment of CAR T-cell administration momentous and significant, others felt it was underwhelming and detracted from the treatment experience.

### Support person experience

Our interviews identified the crucial role that support persons played in the care of patients receiving CAR T-cells. This included assistance with monitoring for CAR T-cell-related adverse events, and co-ordinating the many non-medical challenges that arise as a patient goes through cancer treatment. In addition, support persons often played an active role by advocating for the patient. Reckoning with these new responsibilities, while managing the psychological stress of a close family member's life-threatening illness, presented a significant emotional burden for support persons.

### Bridges

A characteristic of CAR T-cell therapy treatment for New Zealanders was the need for external organisations to serve as 'bridges' to cover gaps in patient care that currently exist in the funded health system. These 'bridges' include charitable organisations, such as Leukaemia and Blood Cancer NZ, who provide financial assistance for travel and accommodation costs during treatment, psychosocial support and participation in support groups for patients to meet other patients receiving similar treatment. Such support groups also arise organically, such as via social media. Community hospices also provide a valuable service providing both end-of-life care and community outreach palliative care input.

The current healthcare system provides a National Travel Assistance programme, which covers a portion of the logistical and accommodation costs for patients and support persons travelling out-of-region for medical treatment. However, the gaps in coverage of this programme result in costs falling to either patients or to charitable organisations.

## Global theme 3: golden opportunities: reimagining CAR T service delivery

The final major theme related to improvement opportunities for a future national CAR T-cell therapy service.

### Centralised point of information

It was recognised that the introduction of a CAR T-cell therapy should be accompanied by education and support structures to help clinicians and patients unfamiliar with the therapy to access it. For clinicians, this would require visible and accessible specialists to liaise with, and clear and accessible guidance of the referral process. For patients, this involves quality education materials to explain the treatment in understandable terms, preferably in video form in addition to text.

### Health navigator

It was identified that a health navigator (sometimes referred to internationally as a 'concierge') could serve a role as a patient liaison throughout treatment, coordinating both the medical and non-medical aspects of care (including logistic, financial and psychosocial considerations), providing patient advocacy and education about aspects of treatment. An idiosyncratic element of CAR T-cell therapy is the cell manufacturing period, leading to a period of uncertainty for patients and referring clinicians as the outcome of CAR T-cell manufacture is awaited. Both clinicians and patients identified a potential role for a 'cell tracker', with which the progress of cell manufacture could be tracked over time, to manage expectations and provide reassurance.

### Changes to treatment delivery

Several potential alterations to current CAR T-cell therapy clinical delivery were identified. First was the possibility of conducting a greater proportion (or even the entirety) of monitoring as an outpatient following cell infusion. This was felt to offer benefits for patient well-being as well as for health resourcing. Second, to address the underwhelming nature of cell administration, a ceremony for CAR T-cell administration was suggested, to imbue the procedure with a reverence befitting the occasion. This could involve family, friends or referring clinicians, either in person or via videoconference, as well as additional cultural or religious elements such as presence of kaumatua (Māori elders) and a karakia (blessing).

### Dedicated clinical staff

It was identified that a CAR T-cell therapy service requires dedicated resourcing and training, rather than unfunded incorporation into overstretched bone marrow transplantation services. Treatment hubs should have dedicated CAR T-cell staff to allow a consolidation of skills and experience, and to ensure safe treatment delivery.

## Indigenous sovereignty

Finally, it was identified that a future CAR T-cell service should include provisions to respect indigenous sovereignty over data and cells. For data, this should include anonymising or deidentification of data shared beyond treating clinicians (unless explicit consent is given), as well as the understanding that data would lead to improvements for the patient group that the data is sourced from. For cells, this would involve explanation of where cells would be located, what processes would be undertaken with them and what would happen to remaining cells that are not required for their cellular therapy. As an example, one Māori participant underwent a cell collection procedure in Aotearoa NZ, before undergoing cell collection for a commercial CAR T-cell therapy overseas. They highlighted that in Aotearoa NZ they were informed about location of their cells during manufacture and were given the option to either donate or dispose of cells that were not required for treatment. Conversely, during their treatment overseas, they were not informed of the location of their cells or the manufacturing progress, nor given options about fate of their cells.

## Codesign workshop

Participants noted that, for patients and support persons attempting to access CAR T-cell therapy, many of the challenges to be navigated are at or before the start of the pathway, and suggested refinements to the prereferral process, and monitoring of referrals to ensure access is equitable.

Participants commented that, although many elements of the current pathway could be improved on, including consent processes, support services and cultural elements, the primary focus of patients and support persons is to access CAR T-cell therapy, and that the need to improve these areas should not prevent initial adoption of the therapy.

Participants were asked to comment on which of the potential improvements (broadly identified in global theme 3) they would prioritise. This resulted in strong support for the health navigator role (to assist patients and families with non-medical and medical navigation of the health service and the effect that cancer treatment would have on their lives); the need for culturally appropriate care for all patients; and a centralised dashboard for information (to provide information about how to access CAR T-cell therapy and what treatment would entail). Elements such as a tracker for cells after extraction, and a ceremony around cell infusion, were considered improvements but less crucial to the success of a CAR T-cell therapy programme.

## DISCUSSION

Aotearoa NZ has an unmet need for funded CAR T-cell therapies for B-cell non-Hodgkin lymphomas and other B-cell malignancies. As we expect CAR T-cell therapy to become a standard of care in the future, it is important that consideration is given to the perspectives of patients, support persons, clinicians and administrators, in the design of a national service. Therefore, we performed this qualitative study to generate insights into the current experiences of CAR T-cell therapy, and the areas which could be improved on.

Elements of the current service were praised by participants. First, CAR T-cell therapy was considered more tolerable than other treatments such as chemotherapy and autologous stem cell transplantation. Second, the role of charities and other organisations in expanding on the support provided by healthcare providers, and mitigating the challenges facing patients and support persons through treatment was applauded. Third, the use of strategies to respect tissue sovereignty within the ENABLE trial was thought to lay important groundwork for further research in this area in future.

We identified several areas for potential improvement of CAR T-cell delivery. First were strategies to improve communication with patients and referring clinicians, and therefore, minimise some of the barriers to access identified in table 1. Some patients had to ask to be referred for the CAR T-cell clinical trial, suggesting that a more robust system of informing clinicians of open clinical trials is required. For a standard-of-care service, strategies could include a centralised 'dashboard' for new CAR T referrals (with clear guidance for clinicians on eligibility, process and time frames), and high-quality educational materials for patients and family in understandable language. Communication could also be improved with a 'tracker' of cell collection and manufacture, which would provide patients and referring clinicians with an insight into time frames.

Other improvements to improve the treatment experience included a transition to outpatient management, which would result in benefits for patients (improved psychosocial well-being and avoidance of infection while in hospital), alongside resourcing benefits (reduced inpatient burden and less overall staff input). Such a transition would require more robust assistance from a National Travel Assistance programme, to ensure that treatment remains a realistic option for patients with fewer resources. A ceremony for CAR T-cell administration, including involvement of family and referring clinicians, and incorporating cultural or religious elements, may be appreciated by many patients.

Given the ethnic health disparities in Aotearoa NZ, including poorer cancer-specific survival for non-Hodgkin's lymphoma and myeloma among Māori,[10] a national CAR T-cell service will need to be designed with an explicit equity focus to ensure it improves Māori health outcomes and avoids replicating pre-existing issues with access to treatment and informed consent. Given the socioeconomic disparities since colonisation, this may require increased support to access treatment and to mitigate financial consequences for the patient and support persons relocating for treatment. It would also require culturally appropriate care within the CAR T-cell therapy

team. Patient autonomy over data and cells was strongly valued, with the option to decide on how both are used.

Finally, we identified the need for treating hubs with consolidated experience and skills, to ensure that clinicians can deliver this new therapy safely. However, it was felt that keeping CAR T-cell therapy in a single treating centre, even in a country with a relatively small population such as Aotearoa NZ, could perpetuate existing geographical inequities. Therefore, consideration will need to be given in the medium term to balancing the relative trade-offs between efficiencies gained from consolidated care and the benefits to patients of care closer to home.

One of the key strengths of this study is the diverse viewpoints obtained by enrolled participants, providing a '360°' view of the CAR T-cell therapy process in Aotearoa NZ—that is, not only patients, but support persons, clinicians and administrators. We intentionally enrolled patients and support persons with experience of both positive and negative outcomes after CAR T-cell therapy, as well as participants with a range of ages and ethnic backgrounds. This diversity of viewpoint, we feel, improved our ability to obtain a broad range of data on current experience and helped us to identify new solutions.

Another strength of our study is the wide range of topics covered relating to CAR T-cell therapy. The flexibility of semistructured interviews, as well as the wide variety of associations of participants to CAR T-cell therapy, means that this project considered many different facets of the treatment pathway, from the individual (eg, patient and support person experience and communication between clinicians) to the system-level (identifying areas of inequity, and how a future service should be designed). This data-driven approach, as opposed to a theory-driven approach, meant that the insights from participants could extend beyond the areas foreseen by the prespecified interview template.

This study was designed in discussion with Māori, and specifically aimed to recruit Māori participants, to inform ways a CAR T-cell service should be designed to improve Māori health outcomes. Our findings highlight the need for further research regarding tikanga, or Māori custom, relating to CAR T-cell therapies, including the role and form of karakia (blessings) at the time of cell collection, administration and disposal. Our findings regarding the needs of support persons touch on the importance of a holistic approach to health outcome evaluation, which for Māori includes mental, spiritual and family/community well-being, as well as physical health (the Te Whare Tapa Whā model[19]). Such work should be conducted using a kaupapa Māori (Māori-led) approach, to ensure the design of a new health pathway actively addresses inequities in Indigenous health outcomes.

A limitation of this study is that by interviewing a wide group of participants, and having a semistructured interview with scope to follow the interviewee to cover topics in more depth, data saturation was not reached in all areas. Indeed, given the limited numbers of New Zealanders who have received CAR T-cell therapy, data saturation

may not have been possible. A further restriction to research into patient experience of relapsed/refractory non-Hodgkin's lymphoma is that the experience of patients not responding to treatment may be lost due to rapid progression of their disease. While we attempted to ameliorate this by interviewing support persons, clinicians and administrators, this may have served to dilute the patient voice. We attempted to mitigate this by testing the thematic analysis within a participant workshop, to gain reflections on themes raised by others. We also note that the themes we have identified, including barriers to accessing CAR T-cell therapy, align with those identified elsewhere.[3 6 8 13 20] We also acknowledge that applying a deductive approach to thematic analysis may have restricted the reporting of themes among participants. Guiding participants along the treatment pathway by way of discussion was a strategy to assist in reducing distress (patients and whanau could anticipate the next step in questioning), and also to assist in the building of a reimagined pathway. An alternative approach could have been to use an inductive approach, with a broader approach to questioning. This may have resulted in themes that did not necessarily emerge in this work. In fact, during the feedback hui, participants commented that the journey before seeking treatment, and after finishing treatment, was of equal importance to them; however, this was beyond the scope of this project.

The focus of the interviews in this study was on the treatment experience and on areas of focus for future service design. While specific suggestions arose that might be incorporated into a future pathway (see table 1, global theme 3), these suggestions alone are not sufficient to address all of the challenges that were identified, particularly with regard to equity. It is hoped that this work will help to identify areas for future focus; such research could include a more comprehensive identification of strategies to address the challenges raised.

While other studies have assessed the patient experience of CAR T-cell therapy via qualitative interviews,[13 21] or assessed the principles required for a CAR T-cell service,[8] our study is the first to examine this in Aotearoa NZ. This study emphasises the importance of codesign to elevate the voice of the indigenous Māori population, and to address the current severely limited access to CAR T-cell therapy. The suggestions unearthed in this study should help development of an intentionally designed service that results in health gains for Māori, and does not exacerbate health inequities.

In summary, the RE-TELL study has defined the current experience of New Zealanders receiving CAR T-cell therapy; identified some of the barriers to those seeking treatment; and identified improvements for a future service, as well as areas for attention for those designing this service. Although our research is specific to Aotearoa NZ, elements of our findings are likely to apply to other countries intending to develop CAR T-cell therapy services, particularly those countries in which indigenous people experience health inequities.

**Acknowledgements** The authors would like to thank the participants for their time and insights, and for their willingness to share their experiences. We acknowledge those who have passed, and uphold this study as part of their legacy. We also acknowledge Myra Ruka for input into design of the study and review of the manuscript.

**Contributors** RF, OA and RW designed the study and wrote the study protocol, with RF acting as guarantor for study conduct and publication. RF, OA, KK and MJ performed study procedures. RF, OA, DOS and RW analysed the data and drafted the manuscript. All authors reviewed and approved the final manuscript.

**Funding** This study was supported by funding from Janssen-Cilag to Deloitte and the Malaghan Institute (grant: N/A). RW is supported by a Clinical Practitioner Fellowship from the Health Research Council (19/139). RF is supported by the LifeBlood Trust. The Malaghan Institute acknowledges philanthropic funding support from David Downs and The Thompson Family Foundation.

**Competing interests** RF, RW and DOS are employees of the Malaghan Institute of Medical Research, which received funding from Janssen-Cilag towards the cost of this research. OA, KK and MJ are employees of Deloitte, which received funding from Janssen-Cilag through a contract for services. Janssen-Cilag had no involvement in the conduct of the research, data collection, data analysis or in preparation or approval of the manuscript.

**Patient and public involvement** Patients and/or the public were not involved in the design, or conduct, or reporting, or dissemination plans of this research.

**Patient consent for publication** Not applicable.

**Ethics approval** Ethical approval for this study was provided by Victoria University of Wellington Human Ethics Committee (ref: 29982) before any study activities were commenced. All participants gave written informed consent.

**Provenance and peer review** Not commissioned; externally peer reviewed.

**Data availability statement** No data are available. The participants of this study did not give written consent for their data to be shared publicly, so due to the sensitive nature of the research supporting data is not available.

**ORCID iD**
Robert Fyfe http://orcid.org/0000-0002-7635-3350

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
