## [Reviewer comments · BMJ Open]

ARTICLE DETAILS

TITLE (PROVISIONAL)	Experiences and perspectives on chimeric antigen receptor (CAR) T-cell therapy among recipients, carers and referrers (RE-TELL): a qualitative study to inform CAR T-cell service design
AUTHORS	Fyfe, Robert; Anstis, Olivia; Kapadia, Kushant; Jordan, Mallory; Sword, Danielle; Weinkove, Robert

VERSION 1 – REVIEW

REVIEWER	Emma Kirby University of New South Wales
REVIEW RETURNED	07-Jun-2023

GENERAL COMMENTS	This is an interesting article, and speaks to important issues of access and equity in healthcare treatment and experience. The data presented is novel, and thought provoking, and the methodological process is refreshing – particularly in terms of the engagement with participants. My main concern relates to the organisation of thematic results. There is a lack of information about the approach to thematic analysis; relatedly, the global themes derived from this process seem to mirror/directly reflect the domains asked about in the interviews. In this way, it is difficult to see how the global themes have been arrived at through analysis, rather than already decided – these are both ok options; rather, it is the incongruity of approach and results that is somewhat confusing. I detail various points in need of further clarification and explanation below; I have also included some suggestions as a means to be constructive. I hope that the authors can make relatively straightforward revisions that would result in the article being not only appropriate for publication, but a useful contribution to the literature. Specific points: - Abstract: Consider changing how you describe the interview ‘themes’, so as to distinguish from themes derived from analysis. Change to “... interviews focused on domains of...”- P7:37: Suggest moving sentence on future service provision to Discussion or Conclusion sections- Methods: The authors intended to recruit 19 participants – why 19? Why not keep interviewing until saturation is reached? This is not a criticism per se; rather it requires more engagement and explanation. Some justification is necessary to determine/explain how this number might equate to a range of perspectives, and how this number relates to the overall sampling frame, as well as issues of richness of data.- How were participants recruited (ie explain process of sampling and recruitment)?, and from what settings?- The subheading: ‘Statistical and ethical considerations’ should be changed to Ethical consideration, and moved down accordingly.
---

	Details of analytic process, including coding processes, should immediately follow in the description of thematic analysis.  - Related to both Methods and Results: Some description of approach to thematic analysis is needed in relation to the thematic findings, in particular (if to use Braun & Clarke's terms) whether inductive, abductive, or more theoretical/deductive. The issue here is that the current description implies inductive (eg codes related to the significance of issues mentioned), but the themes derived seem deductive (eg related to equity, access, issues already identified as focus areas). Some clarification is needed. - There is some slippage in the results between reporting of thematic findings and potential implications of these data. For example: P15:618: this sentence, and others like it in the results and elsewhere, should be moved to the Discussion (unless, the participants themselves discussed or shared their view that without mitigation strategies, this could present greater access barriers to Māori and Pacific peoples?) - Results: Theme 2; perhaps the authors might consider changing Current experiences to Lived experiences (at present, 'current' implies participants are talking about issues they are experiencing currently, rather than reflections on the lived experience). That said, Experiences through Treatment is a key domain of the interview – more emphasis is needed on justification of process: how was this theme derived from analysis, rather than reflective of questions asked? - Theme 3: Again, better explanation of the process of deriving this theme from thematic analysis is needed, given that the Global theme is 'Future Improvements', and the Interview Guide has a section on 'Treatment aspects that could be improved upon'. This seems more of a domain of questioning rather than a derived theme? - The authors state: ...'we performed this qualitative study to generate insights into the current experiences of CAR T-cell therapy, and the areas which could be improved upon.' The revised manuscript needs to attend to how themes were generated through thematic analysis of these experiences and suggestions for improvement. The experiences and suggestions themselves cannot really be derived themes, as they represent the aims and questions set out in the study. Themes could be (just as an example) 'Navigating opaque health systems and pathways' – this might account thematically for some of the barriers to access, and help illuminate equity issues (through for example learning about treatment options through word of mouth). Another example might be 'Uncertainty, fear and hope', which the authors do a good job of describing (both in theme 2 and in the supplementary pathway diagram. I'm not suggesting that these should be themes – I'm trying to give some examples of themes that might be arrived at through thematic analysis that speak to the issues raised, but without replicating the interview guide domains.
--	---

VERSION 1 – AUTHOR RESPONSE

- We have provided clarification regarding our methodological approach in the Methods as suggested, and have made further comment in the Discussion on the limitations this approach has. Specifically, we have identified our approach as deductive, and the specific process followed (in line with a published methodology from Braun and Clarke, cited in the references). We have acknowledged that a deductive approach may have restricted the reporting of themes for participants, as compared with an inductive approach, and commented on the reasons for the choice of methodology.
- We have shifted sentences in both the Introduction and Results section that involved interpretation of the data to the Discussion section, as suggested.

- We have provided clarification in the Methods section regarding the justification of our sample size. This was arrived at due to a combination of factors. Firstly, the scarcity of the therapy in New Zealand and the realities of the high mortality in post-therapy relapse mean that we identified only 11 eligible patients. We intended to recruit 6 patients which we felt would provide a reasonable overview of current experience of patients, but due to either illness or logistical issues were only able to recruit 5. We then attempted to recruit approximately 5 each support persons, clinicians and administrators, in order to balance a desire to explore additional perspectives and experiences with not wishing to overshadow the patient experience.
- We have clarified in the Methods the sampling and recruitment process. Participants were identified through a range of channels: involvement with a CAR T-cell clinical trial programme; informal patient/support person groups; and involvement with charitable organisations. They were recruited through e-mail contact by the primary investigator.
- We have considered the reviewer’s insightful comments on the naming of global themes, and took the opportunity to review and reconstruct these names to speak more directly to the issues raised by participants involved. We thank the reviewer for drawing our attention to this, and giving us the opportunity to craft a manuscript that more truly represents the data shared by our participants. Specifically, we have renamed these to:
 - o Global theme 1: Sociocultural factors impact CAR T access
 - o Global theme 2: Varying emotions, roles and enablers. (This was chosen to include recognition of both the emotional tumult of the treatment experience, but also the elements that lead to patients arriving at, and making it through, treatment).
 - o Global theme 3: Golden opportunities: re-imagining CAR T service delivery (This was chose to acknowledge the broad sentiment amongst participants that this current moment is a chance to re-consider how the treatment is delivered, before it is an ingrained reality (rather than, for example, a desire to simply proceed apace with treatment delivery with all available haste). It but also reflects a specific quote from a participant, that the treatment is a “spoonful of gold”).
- We have also reformatted aspects of the manuscript layout and naming, and corrected specific errors that were noted in the manuscript:
 - o The title “Statistical and Ethical Considerations” was changed to Ethical Considerations, and the description of the analytic process that followed has been moved to follow the thematic analysis.
 - o The title of the manuscript has been revised to include the research question/aim, study design and setting.
 - o The abbreviated study name is now used in the abstract and main text, as suggested.
 - o The heading “Patient involvement” has been amended to “Patient and Public involvement.
 - o The two results sections have been combined, with subheadings used to separate this into two sections.

VERSION 2 – REVIEW

REVIEWER	Emma Kirby University of New South Wales
REVIEW RETURNED	14-Nov-2023
GENERAL COMMENTS	The authors have engaged with the reviewer comments in a considered and thorough engagement. I hope that the article is well-received once published.